# Sustainable Development of the Arctic Indigenous Communities: The Approach to Projects Optimization of Mining Company

**Andrey Novoselov** [1], **Ivan Potravny** [2], **Irina Novoselova** [3] **and Violetta Gassiy** [4,*]

[1] Department of Mathematical Methods in Economics, Plekhanov Russian University of Economics, 36 Stremyanny Lane, 117997 Moscow, Russia; alnov2004@yandex.ru

[2] Basic department of Project and Program Management Capital Group, Plekhanov Russian University of Economics, 36 Stremyanny Lane, 117997 Moscow, Russia; ecoaudit@bk.ru

[3] Department of Integrated International Ecological Problems and Wildlife Management, Moscow State Institute of International Relations (MGIMO University), 76 Prosp. Vernadsky, 119454 Moscow, Russia; iunov2010@yandex.ru

[4] Department of Public administration, Kuban State University, 149 Stavroposkaya, 350040 Krasnodar, Russia

\* Correspondence: vgassiy@mail.ru; Tel.: +7-861-235-35-92

**Abstract:** The article discusses the issues of sustainable development of indigenous communities in the Arctic based on the optimization of projects of mining companies. The purpose of the article is to develop tools for decision-making to optimize the mining projects based on economic and mathematical models. The authors suppose that, by comparing and selecting different options for resource extraction, the use of various technologies and the impact of projects, the conditions of the traditional life of indigenous peoples, the preservation of health, it is possible to find a compromise solution for stakeholders. The case-study of Alrosa—a diamond giant mining in Yakutia is researched in the paper. To ensure sustainable development of traditional lands, it is proposed to optimize mining projects, in order to carry out a project maneuver during Arctic development. The project maneuver of the mining company makes it possible to choose the optimal solution from the existing alternatives for the extraction of minerals. The authors propose criteria and procedures for the selection of alternative options for the implementation of extractive projects. The alternative projects selected in this way make it possible to compensate to indigenous communities for the negative impact during industrial development of the Arctic.

**Keywords:** project; mining company; minerals; indigenous peoples; the Russian Arctic; criteria of interests; alternatives; compensation for negative impact; optimization model; compromise

## 1. Introduction

Currently, to ensure socio-economic development and to improve the quality of life of the local population in the Russian Arctic, a number of projects are being implemented for the extraction of minerals, transport, and social infrastructure development [1].

It is obvious that projects on exploration and extraction of minerals in the Arctic can not only contribute to the involvement of natural resources in the economic circulation, generate income, and create new jobs for the local population but also be accompanied by a negative impact on the environment, traditional lands, climate, and health of local residents.

In 2020, Russia has adopted Arctic Strategy for the period up to 2035. The Strategy determines the peculiarities of the Arctic zone. It also fixes the special approaches to its socio-economic

development and national security. The Strategy pays attention to the high sensitivity of the Arctic ecological systems, the indigenous peoples, and external influences, as well as climatic changes that create both new economic opportunities and risks in the field of economic activity and environmental protection. It should be noted that climate change and industrial development in the Arctic region make more risks for indigenous peoples than other factors. For example, the extraction of minerals, in particular diamonds, along with melting permafrost increase the risks for human activity in the Arctic.

Available studies show that the impact of global warming in the Russian Arctic is stronger than in other regions with low and middle latitudes. The climate change affects the water and carbon cycles. For example, Huan J. [2] and Suzuki et al. [3] pointed to the expansion of drylands in the highlands of the Russian Arctic.

The sociological surveys of indigenous peoples in Yakutia show that they link bad quality of water resources with climate change and active mining of the tributaries of the river. In particular, the level of water and banks of Anabar river, which flows into the Laptev Sea, collapse due to thawing of permafrost and active diamond mining [4].

Development of alluvial diamond mining, for example, in the Ebelyakh river basin, a tributary of the Anabar, may have an impact on hydrological changes. In addition, the extraction of diamonds from the channels, especially in the places where the settlements of indigenous peoples are located, can create a problem with the provision of drinking water, with following negative impact on quality of life. It is interesting to note that in the social polls of the Arctic population on the impact of the Covid-19 pandemic on the life of the region, many respondents noted that a decrease of the economic activity of mining companies resulted in the improvement of the environmental situation [5].

Modern research shows a fairly close dependence of mining activities with climatic changes [6]. As a rule the extraction projects take into account the issues of obtaining the actual income from the development of deposits. But there is a need to consider the issues of their impact on the living conditions of indigenous peoples, the preservation of their traditional environment, and on water resources and climate to ensure sustainable development of local communities [7].

Currently, in Anabar region, a so-called "Diamond province" of Yakutia, a mining giant Alrosa envisages the implementation of several diamond mining projects. These projects have several alternative options, and some of them could be implemented without fail, while the other part can be implemented if it is economically feasible. Alternative options for their implementation imply the development of various deposits at different facilities, the use of various technologies, etc. [8]. As a result, depending on the alternative, the project may affect different areas; have different impact on the recipients, primarily on the indigenous peoples and their traditional lands, as well as on the environment, water objects, and climate situation.

Currently, two main technologies are used in diamond mining: a diamond mining from kimberlite pipes (root deposit) and alluvial diamond mining. For example, the Verkhne-Munskoye diamond deposit belongs to the first type of project technology, and alluvial diamond deposits in the river basins of Bolshaya Kuonamka and Talakhtakh are realized by the second type in Anabar region.

Not depending on the type of using technology, the impact of a diamond mining projects will include the change in livelihoods of the indigenous communities, in the landscape, alteration, and disruption of river channels, waste generation, a decrease in the resource productivity of traditional lands (reindeer husbandry, hunting, fishing, gathering of wild plants), etc. [9]. In these conditions, it is necessary to choose a set of ongoing projects that allows for the cost-effective mining of minerals, in order to implement measures to support the local communities and the socio-economic development of the territory in the interests of the indigenous peoples of the Arctic.

## 2. Materials and Methods

In search of a new growth, in 2011, the mining company Alrosa began exploration at 5 new deposits in western Yakutia, three of which are primary (Verkhne-Munskoye, Mayskoye, Dalnyaya pipe) and two placer deposits (Ebelyakh and Gusinaya). The Verkhe-Munskoye field was discovered in 2007 in Olenek region, western part of the Diamond province. It is located in the north of the

Yakutia, 170 km from the town of Udachny. This deposit can produce about 1.8 million carats of diamonds per year, and its reserves are sufficient to continue mining for more than 20 years—until 2042. It will support the company's stable diamond production by providing jobs for local residents. In total, it provides for more than 800 new jobs for local residents. Already today, about 20% of workers at the field are local residents. It should be noted that the new deposit will compensate for the losses of the company and Yakutia due to the closure of the Mirniy mine, which was flooded with water in 2017. Revenues to the regional budget for the entire period of development of the deposit will amount to $3 billion. Table 1 shows the characteristics of the company's deposits in terms of their reserves and the cost of market products.

**Table 1.** Characteristics of the Alrosa deposits: the reserves and the cost of market products.

| Diamond Fields | Diamond Reserves, Million Carats | Value of Goods, $ Million |
|---|---|---|
| Verkhne-Munskoye | 40.0 | 4300.0 |
| Maiskoye | 13.3 | 930.0 |
| Dalnyaya | 10.2 | 600.0 |
| Ebelyakh | 25.1 | 900.0 |
| Gusinaya | 3.4 | 120.0 |
| Total | 92.0 | 6850.0 |

Resource: New diamond fields explored by Alrosa may turn into possible sources of reserve growth (see online at: https://www.interfax.ru/business/235642 (accessed 19 February 2020).

The territory of the Verkhne-Munskoye field is located on the right bank of the Muna River, the left tributary of the Lena River—one of the largest in Siberia. The territory is located in the municipal area "Olenek Evenk national district" of Yakutia. In 2011, according to special federal order, a part from natural reserve of the Olenek was withdrawn for exploration and production of diamonds. Figure 1 shows the Diamond province of the Republic of Sakha (Yakutia) with the location of the considered diamond mining fields. The area of economic activity is 790.8 ha for open pit mining of four diamond pipes. Overburden stripping works are carried out using the explosive method. The researched territory of the Olenek Evenk national district belongs to the category of traditional lands. There are all types of the traditional economic activity of the indigenous peoples in the area: reindeer herding, fishing, hunting, and gathering wild plants and berries.

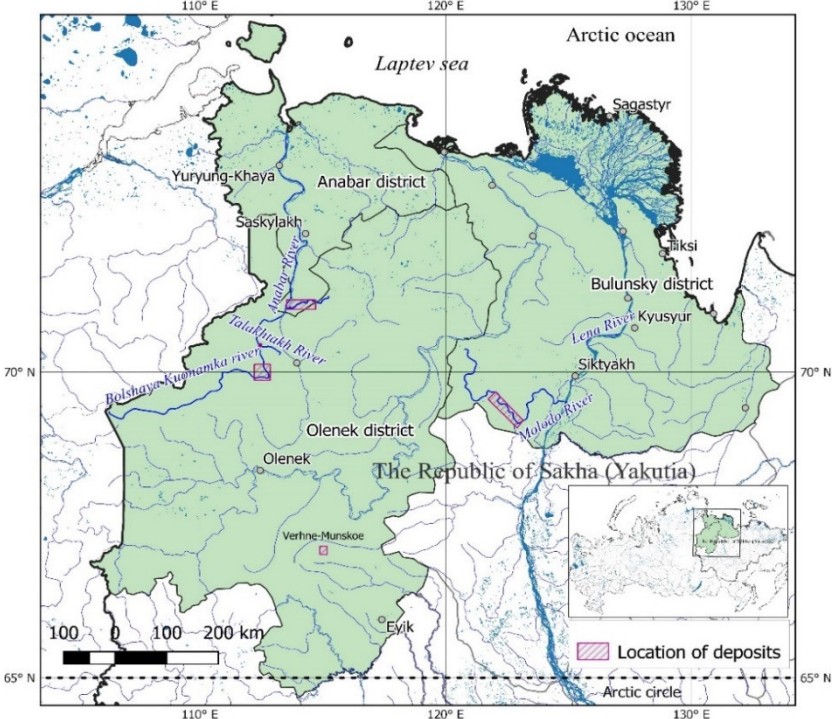

**Figure 1.** "Diamond Province" of the Republic of Sakha (Yakutia).

The methodology of this study is based on the concept of benefit sharing during industrial development of the Arctic [7]. At the same time, ensuring sustainable development and obtaining joint benefits by the mining company and indigenous peoples means that investment projects are selected and approved taking into account the interests of all stakeholders [8]. In fact, such processes are the result of projects' optimization of extractive companies based on economic, environmental, ethnological, and social criteria.

According to current Yakutia legislation, the mining in traditional lands supposes the impact assessment or "ethnological expertise" [9]. It is associated with the assessment and compensation of possible damage to the indigenous peoples in the zone of project impact [10,11]. In the paper, the authors consider the Ebelyakh project on alluvial diamond in Anabar region which impact assessment they made during field research in 2015–2016. According to the results of the ethnological expertise, it was recommended to conclude an Agreement on social and economic development of the region. Such agreement was signed between mining company and the municipal district. It appears as a mechanism for sustainable development and adaptation of indigenous peoples to modern conditions. The Agreement includes the following points:

– the financial support of alternative forms of economic activity;
– the preservation of traditional culture (customs, rituals and national holidays); and
– the local labor market development (including the employment of the local population and its education); and
– the financing of programs for indigenous youth development.

The researched project on alluvial diamonds extraction is planned to realize until 2025. It is located in the traditional lands of Anabar dolgan-evenk national district with Saskylakh village as the capital. The Anabar river with the tributary Ebelyakh belong to the basin of the Laptev Sea. The Ebelyakh River's diamond placer is the largest in Russia and one of the largest in the world. Figure 2 shows the location of the mining area along the Ebellyakh River. This deposit has an impact on the territories of traditional lands of the indigenous peoples. According to the results of the ethnological expertise, the losses to indigenous peoples during the industrial development of the area are evaluated to 4120.0 thousand rubles per year (55,000US$).

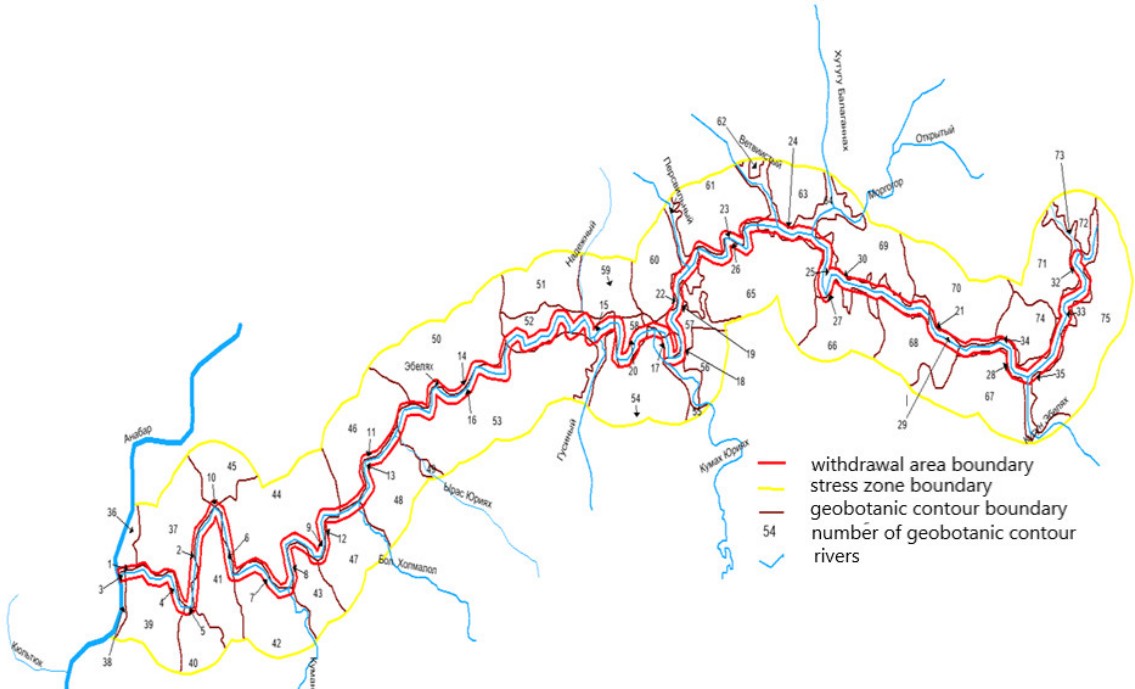

**Figure 2.** The location of the extraction of alluvial diamonds on the Ebelyakh river.

In general terms, the researched problem could be formulated as follows. The mining company has a number of projects j, some of which are mandatory, and the rest can be implemented if it is economically feasible. For each of the projects j, a set of alternative options $J_j$ for its implementation are set, which differ in terms of geographic location, investment volumes, cash flow, duration (or timing) of implementation. Of this multitude of alternatives, a single option $i \in J_j$ must be chosen. To eliminate conflicts with the local population, the mining company implements compensation projects [12] aimed at improving the living conditions of the local residents, reducing environmental pollution, supplying the population with necessary goods, providing medical care, etc. For each alternative project of the mining company, it is proposed to carry out a set of compensation projects [6], which requires the appropriate amount of funding and is regulated by a variety $P_{ij}$ (Table 2).

An alternative option for the development of minerals refers to options for implementing a project in different territories or using different technologies. Transferring the project from one alternative to another will reduce the negative impact on the recipients, including the impact on traditional activities and the indigenous communities, and change the scale and duration of the company's mining operations, as well as the volume of mining operations.

Compensation projects, here, mean projects aimed at supporting indigenous peoples and their traditional crafts, ensuring their sustainable development, ecological rehabilitation of territories affected by resource extraction, including monetary compensation to the population, providing the population with medical assistance and communications, computerization and training of the population; and supplying the population with food, fuel and lubricants, and fishing and hunting accessories, as well as vehicles (ATVs, snowmobiles, motor boats, etc).

**Table 2.** The structure of the projects of the mining company, which are objects for the analysis.

| Perspective Projects of the Diamond Mining Company | |
|---|---|
| Variety of projects, which can be applied for implementation $j = 1, 2, \ldots n$ | Variety of projects which are obligatory for fulfillment $j = n + 1, n + 2, \ldots m$ |
| Project alternatives which can be applied for implementation $i \in J_j$, $j = 1, 2, \ldots n$ | Project alternatives which are obligatory for fulfilment $i \in J_j$, $j = n + 1, n + 2, \ldots m$ |
| Sets of compensatory projects $p \in P_{ij}$ for each of the alternatives of the mining projects implementation $i \in J_j$, $j = 1, 2, \ldots m$ | |

At the same time, this does not mean that the alternative with a minimum amount of financing for compensation projects is the most preferable for a mining company. The options that are more costly in terms of compensation volume can allow obtaining a larger volume of mineral production, requiring a smaller amount of capital and operating costs. Thus, the mining company can perform a project maneuver that allows the optimal solution to be selected from alternative mining solutions. Hence, it becomes necessary to consistently solve two logically and informationally related tasks:

- Task 1—determining the priority of compensation projects that are supposed to be implemented in case of choosing the appropriate alternative option for the implementation of the extraction project;
- Task 2—identifying a set of mining projects that perform the economic interest to the mining company and provide an opportunity to implement associated compensation projects to meet the interests of the local population. This activity is aimed to eliminate the conflict of interests and strengthening the company's reputation.

The solution of the tasks is reflected in the enlarged block diagram (Table 3), where blocks 1 and 2 correspond to Task 1; blocks 3 and 4—to Task 2. From the block diagram, it can be seen that the solution to task 1 is based on expert information received from the local population. The information makes it possible to find out the importance of the proposed offset projects carried out as the specific alternative option. Blocks 3 and 4 in this diagram correspond to the second problem-solving. During block 2 implementation, the model for determining the optimal alternative options for mining projects realization is filled with numerical characteristics: the costs and the expected profit; for offset projects—the cost of their implementation and priority.

**Table 3.** The enlarged block diagram of the solution of the assigned tasks.

| Input Information | Information Analysis | Output Information |
|---|---|---|
| Information from local population | 1. Assessment of the priority of social groups | Priority of the social groups |
| Social surveys for the assessment of compensation projects' priority | 2. Social surveys analysis and determination of the priority of the compensation projects | Priority of the compensation projects for each alternative option of mining |
| Characteristics of the mining projects and their alternatives of implementation, compensation projects and their priority | 3. Formation of the selection model of optimum options of the mining projects realization with the priority of compensation projects according to local communities | Numbered model of the selection of optimum options for the mining projects realization |
| Formation of the selection model of optimum options of the mining projects realization | 4. Formation of the optimum option for the mining projects realization based on the local communities' opinion and needs | Determination of the optimum set of the options for the mining projects realization and contributed compensation projects |

Accumulated profit from the implementation of mining projects is calculated through discounting the time functioning of the field, i.e., net present value (NPV). During problem-solving 1, expert assessments cover different social groups of the local communities: reindeer herders, hunters, fishermen, mammoth tusk collectors, etc. From the standpoint of these social groups, it is necessary to assess the priority of the criteria for evaluating compensation projects. For example, the following criteria can be used:

- receiving monetary compensation;
- preservation of traditional crafts, hunting grounds;
- diversification of local production;
- improving health care;
- direct provision of economic and social benefits; and
- preservation of culture, language, ethnic group of indigenous peoples, etc.

Information about the expectations and needs of the local population during project implementation was obtained as the result of a field work of the authors in Arctic Yakutia in 2016–2019 [13].

To solve this kind of priority assessment tasks, it is advisable to use the method of hierarchy analysis by T. Saaty [14]. In this work, we used the author's modification of the hierarchy analysis method, which consists of the following:

Step 1. The priorities of each group are identified according to the number of people by using the formula:

$$W_l = \frac{N_l}{\sum\limits_{l=1}^{L} N_l}$$

(1)

Step 2. Questionnaire survey of each group is conducted to calculate priority criteria. Therefore, it is recommended to use a pared-down rating scale given in Table 4.

**Table 4.** Rating scale of priority criteria.

| Lexical Evaluation | Quantitative Equivalent |
|---|---|
| Priority criteria are of equal importance k and k' | 1 |
| Criterion k is more important than the criterion k' | 3 |
| Criterion k is absolutely more important than the criterion k' | 5 |
| Intermediate values between two neighboring numbers on the scale, when it is difficult to give a precise answer | 2.4 |

The Matrix is filled with reverse symmetric values, i.e., if the value of priority criteria in line k and in column k' equals $a_{k,k'}$, then, in the symmetric matrix square, i.e., in line k' and in column k, value equals $\frac{1}{a_{k,k'}}$. In practice, an evaluation matrix is filled by the representatives of the first group, but also can be filled by individual experts from the analyzed group, and, then, every value of the general matrix is calculated as geometric mean of experts' matrices.

According to the hierarchy analysis method by T. Saaty, the next stage is to calculate the matrix constituency. We find an approximate largest eigenvalue:

$$\lambda_l = \sum_{\substack{s=1 \\ k=s}}^{K} \left( \sum_{k=1}^{K} A_{l,k,s} \right) \left( \sqrt[K]{\prod_{s=1}^{K} A_{l,k,s}} \right)$$

(2)

Then, Consistency Index of the expert assessment is calculated by using the following formula:

$$CI_l = \frac{\lambda_l - K}{K - 1} CS(K)$$

(3)

CS(K) is normative value of the matrix constituency with K—dimensional parameter (Table 5).

**Table 5.** Constituency coefficients CS.

| Matrix Dimension *K* | 3 | 4 | 5 | 6 | 7 |
|---|---|---|---|---|---|
| *CS* | 0.211 | 0.423 | 0.464 | 0.483 | 0.495 |

If Consistency Index is less than 0.1, that means the judgments are perfectly consistent and the results can be used in the further calculations. Otherwise, the expert should be suggested to clarify judgments.

For a consistent matrix of the *l*-group evaluation, the priorities of the criteria are determined with the help of the formula:

$$\alpha_{lk} = \sqrt[K]{\prod_{s=1}^{K} A_{lks}} \qquad (4)$$

Step 3. After completing matrix evaluation criteria with each group of the population, final evaluation of the priorities is carried out using the formula:

$$\beta_k = \sum_{l=1}^{L} W_l \alpha_{lk} \qquad (5)$$

Step 4. Evaluation of the level of a criteria achievement during compensatory projects implementation. Scale of achievement level of established criteria is applied for this evaluations (Table 6). Experts $e$ $(e=1,2,...E)$ assess an impact of a compensatory project $p$ $(p=1,2,...P)$, implemented in the case of $i$ alternative realization of $j$ mining project on a criterion $k$ $(k=1,2,...K)$ using interval evaluation, i.e., they determine an interval $\left(\gamma_{epk}^{\min}; \gamma_{epk}^{\max}\right)$. The resulting value should be found as the arithmetic mean of the obtained interval evaluations:

$$\omega_{pk(ij)} = \frac{\sum_{e=1}^{E}\left(\gamma_{epk(ij)}^{\min} + \gamma_{epk(ij)}^{\max}\right)}{2E} \qquad (6)$$

**Table 6.** Rating scale of the level of criteria achievement during compensatory project implementation.

| Lexical Evaluation of the Level of Criteria Achievement during Compensatory Project Implementation | Interval Evaluation | |
|---|---|---|
| | Min | Max |
| Project is not connected with a criterion | 0.0 | 0.0 |
| Project has an insignificant influence | 0.0 | 0.2 |
| Intermediate option | 0.2 | 0.4 |
| Project has a middling influence | 0.4 | 0.6 |
| Intermediate option | 0.6 | 0.8 |
| Project has a significant influence | 0.8 | 1.0 |
| Project satisfy entirely a criterion | 1.0 | 1.0 |

Step 5. Evaluation of the criteria's satisfaction level during compensatory projects implementation is analyzed with the formula:

$$\mu_{k(ij)} = \begin{cases} 1 & если \ \sum_{h=1}^{P} \omega_{pk(ij)} > 1 \\ \sum_{h=1}^{P} \omega_{pk(ij)} & если \ \sum_{h=1}^{P} \omega_{pk(ij)} \leq 1 \end{cases} \qquad (7)$$

Step 6. Level of criteria's satisfaction during compensatory projects implementation related to $i$ alternative of $j$ mining project is determined using the following formula:

$$\rho_{ij} = \left( \sum_{k=1}^{K} \mu_{k(ij)} \beta_{k(ij)} \right) K^{-1} \tag{8}$$

The solution of Task 2 will help a mining company to perform a maneuver in order to choose options of the mining projects and which will allow to solve two problems at the same time: to gain maximum benefits from mining in the region and to satisfy revealed priorities of the population as good as possible. For this purpose, an optimization model has been developed:

The criterion is maximization of the results from mining

$$f_1(x) = \sum_{j=1}^{n} \sum_{t=1}^{T_j} \left[ \sum_{i \in J_j} \left( V_{jt} - C_{ijt} - Z_{ijt} \right) x_{ij} \right] \left( 1 + r \right)^{1-t} \to \max \tag{9}$$

$V_{jt}$ —volume of realization due to the use of $j$ project per year $t$, *mln euro*;

$C_{ijt}$ —current costs for $j$ project per year $t$, *mln euro*;

$Z_{ijt}$ —capital costs for the development of $j$ project per year $t$, *mln euro*;

$x_{ij}$ —required variable, which takes value 1 if the alternative $i$ for the project $j$ is chosen for the implementation; otherwise—0;

$J_j$ —variety of alternatives for the $j$ $\left( i \in J_j \right)$ project;

$r$ —discount rate *in shares*; and

$T_j$ —project life cycle $j$, *years*.

A criterion of maximization of the population's satisfaction level based on identified priorities (6) is the following:

$$f_2(x) = \sum_{j=1}^{n} \sum_{i \in J_j} \rho_{ij} x_{ij} \to \max \tag{10}$$

Since the extraction projects are divided into two parts (Figure 1), different restrictions are formed for them on the choice of alternative options for their implementation:

- The projects that can be implemented $j = 1, 2, \dots n$ , and each alternative option of implementation can be found or none of the analyzed options can be selected:

$$\sum_{i \in J_j} x_{ij} \leq 1 , \quad j = 1, 2, \dots n \tag{11}$$

- The projects that must be implemented $j = n+1, n+2, \dots m$ , and each alternative option of implementation can be found or none of the analyzed options can be selected:

$$\sum_{i \in J_j} x_{ij} = 1 , \quad j = n+1, n+2, \dots m \tag{12}$$

The volume of investments for the projects implementation is limited $B_t$ for each year $t$, and it is advisable to carry out the calculation on a time horizon covering the entire life cycle of the projects $T_j$ $(j = 1, 2, \dots m)$, i.e., equal to the maximum of the life cycles of mining projects $\max_{j=1,2,\dots m} \{T_j\}$:

$$\sum_{j=1}^{n}\sum_{i\in J_j} K_{ijt} x_{ij} \le B_t \quad t=1,2,\dots \max_{j=1,2,\dots m}\left\{T_j\right\} \tag{13}$$

The resulting model has two optimality criteria (7, 8), therefore, to find a solution, it is necessary to convolve these criteria and search for a solution in the Pareto domain [15]. To agree on the criteria, it is advisable to use a convolution that implements the principle of fair concession from the best values for each of the criteria [16]. The sequence of steps for finding this problem is as follows:

Step 1. Finding the minimum and maximum values of the formed criteria under the given constraints, i.e., solving four problems:

- Maximization of criterion (7) under constraints (9–11) and definition $f_1^{\max}$;

- Minimization of criterion (7) under constraints (9–11) and definition $f_1^{\min}$;

- Maximization of criterion (8) under constraints (9–11) and definition $f_2^{\max}$;

- Minimization of criterion (8) under constraints (9–11) and definition $f_2^{\min}$.

Step 2. Scaling the criteria, i.e., bringing them to the same range of variation (from 0 to 1), the same units of measurement (fractions of deviation from the best value) and the direction of minimizing the deviation from the best value:

$$F_1(x) = \frac{f_1^{\max}-f_1(x)}{f_1^{\max}-f_1^{\min}} \quad \text{and} \quad F_2(x) = \frac{f_2^{\max}-f_2(x)}{f_2^{\max}-f_2^{\min}} \tag{14}$$

Step 3. Determination of the optimal set of production projects from the Pareto compromise area, taking into account a fair concession, based on solving the following problem:

Optimality criterion—the minimum value of the assignment, equal to the criteria under consideration:

$$z \to \min \tag{15}$$

Limitation on the amount of the assignment:

$$z \ge \omega_1 F_1(x) \tag{16}$$

$$z \ge \omega_2 F_2(x) \tag{17}$$

where $\omega_1\ (\omega_2)$ are the weights of the criteria that must satisfy the condition $\omega_1+\omega_2=1$. If the priority is equal, the weights of the criteria should be set $\omega_1=\omega_2=0.5$.

The model also needs to take into account the constraints (9–11). As a result of problem-solving (13)–(15), (9)–(11), the optimal design maneuver of the mining company will be found. Such set of alternative options for the project's implementation supposes the possibility of the deviations from the best values of the criteria (7), (8) but within achievable limits. It should be noted that it is possible to determine the project maneuver for different weights of the optimality criteria $\omega_1$ and $\omega_2$. This will lead to an increase in one optimality criterion by reducing the other.

## 3. Results

The proposed methods and models were tested on the cases of Alrosa mining company and its subsidiary Almazy Anabara which specializes on alluvial diamonds extraction. They mine in the area of 3 rivers: Ebelyakh, Malaya, and Bolshaya Kuonamka. The mining company plans the

implementation of four diamond mining projects. The projects 1 and 2 can be implemented in case of sufficient residual financing and economic feasibility, and projects 3 and 4 must be implemented without fail.

As alternative options for mining the following solutions may be considered [17]:

1.　The extraction of alluvial gold and platinum from previously accumulated mining wastes formed in the results of past economic activities. During 2018–2019, 200 kg of gold and 80 kg of platinum were mined from the waste of the enrichment plant.
2.　Application of technologies for the associated extraction of placer gold during the processing of waste accumulated in sludge ponds and sedimentation tanks during diamond mining.
3.　Restrictions on the mining of alluvial diamonds on the Malaya Kuonamka river near the indigenous community Zhilinda. The river water is the one source of the life support for local residents. The indigenous peoples use it as a drinking water, fishing, and transport communications. Such a wish was expressed by the local population during public hearings on this project.
4.　Maintaining diamond production at the Verkhne-Munskoye deposit as compensation measures for the disposal of the "Mir" deposit as a result of its flooding.
5.　Further expansion of exploration and extraction of alluvial diamonds on 3 rivers.

During projects implementation, it is permitted to perform a maneuver by choosing one of the alternative options (Table 7).

**Table 7.** Extraction projects of Alrosa and their alternatives for maneuver.

| Projects | Compulsory Implementation | Alternatives of Mining Projects Implementation | Places of Projects Implementation and Mining Technology |
|---|---|---|---|
| 1 | In the case of sufficient residual funding and economic viability | 1 | Gusinaya alluvial deposit |
|  |  | 2 | Production and recovering alluvial gold and platinum from the accumulated waste in the concentrating factory on the Ebelyakh's tributaries |
|  |  | 3 | Production and recovering alluvial gold and platinum as byproducts accumulated at the sludge collectors and settling tank in the Molodo and Mayat River |
| 2 |  | 4 | Expansion of search and extraction of alluvial diamonds on the Ebelyakh's tributaries and on the territory of the Molodo River |
|  |  | 5 | Rehabilitation of the Mir (underground diamond mine) |
| 3 | It is obliged to be implemented | 6 | Verkhne-Munskoye underground diamond mining |
|  |  | 7 | Alluvial diamond placer in the Ebelyakh River Basin |
| 4 |  | 8 | Alluvial deposit in the Bolshaya Kuonamka River and Talakhtakh Creek |
|  |  | 9 | Alluvial diamond placer in the Malaya Kuonamka River |
|  |  | 10 | Alluvial gold and platinum mining extracted from accumulated diamond mine waste |

Alternatives in mining projects involves the fulfillment of a set of compensatory projects which include: signing an agreement between a subsoil user and the indigenous peoples on socioeconomic development of the territory, compensation for possible damage caused to the local population in the area of projects implementation and special purpose payments, creation of petrol, oil and lubricants provisioning centers for the local population, supplying with vehicles (snowmobiles, quad bikes, boats), creation of health centers, the procurement and distribution of local products, the equipment provision for local industry (leather production, collecting of mammoth tusk, etc.), reduction of the environmental impact, and noise and vibration caused by mining [18]. These projects are estimated in relation to the set of criteria which is achieved. They include monetary compensation, the preservation of traditional lands, hunting grounds, development of local production and traditional economic activities, improvement of health care, preservation of cultural objects, and other support of local culture. The local population, which is interested in the realization of all above criteria, was involved in the expert assessment. Meanwhile, people were divided into groups: reindeer herders, hunters, fishermen, mammoth, mushrooms, berries, herbs collectors, and other traditional types of activity. The total number of the population inhabited the areas of these projects' implementation is amounted to 4185.

Table 8. shows the calculation of the groups' weight in accordance with step 1 of task 1.

**Table 8.** Calculation of the weight for each group of the indigenous peoples.

| Group Number $l$ | Group Name | Size of the Population Group $N_l$, People | Group's Weight $W_l$, Rate |
|:---:|:---:|:---:|:---:|
| 1 | Reindeer herders | 1300 | 0.338 |
| 2 | Hunters | 810 | 0.211 |
| 3 | Fishermen | 400 | 0.104 |
| 4 | Collectors of mammoth tusks, mushrooms, berries, herbs | 90 | 0.023 |
| 5 | Gatherers of mammoth tusks and bones | 90 | 0.023 |
| 6 | The population engaged in the production of souvenirs, household utensils, processing of products of traditional crafts | 320 | 0.083 |
| 7 | The population engaged in the organization of tourism, leisure, service of traditional crafts | 390 | 0.101 |
| 8 | The population employed in the public sector (government, schools, hospitals, post office, etc.) | 445 | 0.116 |

Step 2 includes the assessment of priority criteria. A comparative assessment is undertaken for each population group based on the expertise with the use of the scale Table 1, and the priority criteria are determined by Formula (4) as it is shown in Table 9.

**Table 9.** An example of the priority criteria assessment for "Reindeer herders".

| Criterion Number $k$ | Criteria | Criteria Numbers $k$ = 1, 2, …, 6 | | | | | | Criterion Priority $\alpha_{lk}$, Rate |
|:---:|:---:|:---:|:---:|:---:|:---:|:---:|:---:|:---:|
| | | 1 | 2 | 3 | 4 | 5 | 6 | |
| 1 | Monetary compensation | 1 | 1 | 3 | 2 | 3 | 3 | 0.26 |
| 2 | Preservation of the territories of traditional natural resource use, hunting grounds | 1 | 1 | 5 | 5 | 5 | 3 | 0.36 |

| | | | | | | | | | |
|---|---|---|---|---|---|---|---|---|---|
| 3 | Development of local production and traditional economic activities | 0.33 | 0.2 | 1 | 1 | 0.33 | 0.5 | | 0.06 |
| 4 | Improvement of health care | 0.5 | 0.2 | 1 | 1 | 0.25 | 0.33 | | 0.06 |
| 5 | Supplying the local population with vehicles, transport infrastructure development | 0.33 | 0.2 | 3 | 4 | 1 | 0.33 | | 0.11 |
| 6 | Preservation of culture, language, ethnos, indigenous peoples | 0.33 | 0.33 | 2 | 3 | 3 | 1 | | 0.15 |
| | Amount | 3.5 | 2.93 | 15 | 16 | 12.58 | 8.17 | | 1 |

Table 10 shows the calculated data and assessment of priority criteria obtained by the Formula (5).

**Table 10.** The priority criteria assessment in accordance with the interest of all population groups.

| Criteria $k$ | Priority Criteria for Population Groups $\alpha_{lk}$, Rate | | | | | | | | Priority $\beta_k$, Rate |
|---|---|---|---|---|---|---|---|---|---|
| | Reindeer Herders | Hunters | Fishermen | Collectors of Mammoth Tusks, Mushrooms, Berries, Herbs | Gatherers of Mammoth Tusk and Bones | The Population Engaged in the Production of Souvenirs, Household Utensils, Processing of Products of Traditional Crafts | The Population Engaged in the Organization of Tourism, Leisure, Service of Traditional Crafts | The Population Employed in the Public Sector (Government Bodies, School, Hospital, Post Office, etc.) | |
| 1 | 0.260 | 0.261 | 0.261 | 0.112 | 0.152 | 0.150 | 0.070 | 0.109 | 0.208 |
| 2 | 0.359 | 0.378 | 0.225 | 0.142 | 0.085 | 0.172 | 0.070 | 0.077 | 0.260 |
| 3 | 0.063 | 0.058 | 0.208 | 0.061 | 0.108 | 0.099 | 0.153 | 0.047 | 0.088 |
| 4 | 0.060 | 0.096 | 0.103 | 0.141 | 0.198 | 0.153 | 0.202 | 0.137 | 0.108 |
| 5 | 0.107 | 0.108 | 0.129 | 0.228 | 0.394 | 0.244 | 0.454 | 0.238 | 0.181 |
| 6 | 0.150 | 0.100 | 0.074 | 0.316 | 0.063 | 0.181 | 0.051 | 0.392 | 0.154 |
| Group's weight $W_l$, rate | 0.338 | 0.211 | 0.104 | 0.023 | 0.023 | 0.083 | 0.101 | 0.116 | 1.000 |

Formula (6) with the scale of Table 4 is used to assess the level of progress in achieving the criteria during the compensatory project's implementation for each of the alternatives of mining projects.

The results of such assessment for the compensatory projects of the first option are shown in the Table 11. The last line in this table gives the assessment of achieving each criteria obtained using the Formula (5).

**Table 11.** Assessment of the progress in achieving the goals with the help of compensatory projects related to the first option $i = 1$ of $j = 1$ project implementation.

| Compensatory Projects | Degree of Criteria Achievement $\omega_{pk(1,1)}$, Rate | | | | | |
|---|---|---|---|---|---|---|
| | 1 | 2 | 3 | 4 | 5 | 6 |
| Special purpose payments | 0.45 | 0 | 0 | 0 | 0 | 0 |
| Creation of free petrol stations | 0 | 0 | 0.1 | 0 | 0.9 | 0 |
| Supplying with vehicles, transport infrastructure development | 0 | 0 | 0 | 0.5 | 0.4 | 0 |
| Creation of health centers | 0 | 0 | 0 | 0.3 | 0 | 0 |
| Preservation of culture, language, ethnos | 0 | 0 | 0 | 0 | 0 | 0.3 |
| Organization of the procurement network and distribution of local products | 0 | 0 | 0.45 | 0 | 0 | 0 |
| Provision with the equipment for the local industry | 0 | 0 | 0.8 | 0 | 0 | 0 |
| Reduction of the adverse environmental impact caused by mining | 0 | 0.4 | 0 | 0 | 0 | 0.2 |
| Assessment of criterion achievement | 0.45 | 0.4 | 1 | 0.8 | 1 | 0.5 |

The degree of the local population satisfaction during the compensatory projects implementation of the option $i = 1$ project $j = 1$ is calculated with the help of the Formula (8) based on the results of Tables 8 and 9 and equals $\rho_{1,1} = 0.11$. The assessments of the degree of the local population satisfaction for the rest options of the analyzed mining projects were obtained using the same calculations, and they are presented in Table 12. This table shows capital and current costs of alternatives during mining projects implementation and annual income of these projects. The net present value (NPV) was calculated using the data, development time, exploitation period of each project (8 years), and 10% discount coefficient. These values are presented in the last column of Table 12.

**Table 12.** Numerical values of option of the mining projects implementation.

| Options $i$ | Mining Projects $j$ | Capital Costs, Mln Euro | Current Costs, Mln Euro | Annual Income, Mln Euro | The Degree of the Local Population Satisfaction $\rho_{i,j}$, Rate | Net Present Value (NPV), Mln Euro |
|---|---|---|---|---|---|---|
| 1 | | 12 | 3 | 9 | 0.11 | 17.21 |
| 2 | 1 | 16 | 4 | 9 | 0.40 | 8.34 |
| 3 | | 21 | 4 | 9 | 0.70 | 3.34 |
| 4 | 2 | 24 | 3 | 12 | 0.30 | 19.82 |
| 5 | | 28 | 4 | 12 | 0.80 | 10.95 |
| 6 | 3 | 18 | 5 | 14 | 0.30 | 25.82 |
| 7 | | 27 | 7 | 14 | 0.80 | 7.08 |
| 8 | | 16 | 4 | 16 | 0.20 | 42.42 |
| 9 | 4 | 19 | 6 | 16 | 0.50 | 29.68 |
| 10 | | 25 | 8 | 16 | 0.80 | 13.95 |

To solve the task of finding an optimal project maneuver, let us make a numeric form of optimization problem based on the data in the Table 12. The first criterion is maximization of the total net present value (NPV) during projects implementation:

$$f_1(x) = 17.21x_{1.1} + 8.34x_{2.1} + 3.34x_{3.1} + 19.82x_{4.2} + 10.95x_{5.2} + 25.82x_{6.3} +$$

$$+25.82x_{6.3} + 7.08x_{7.3} + 42.42x_{8.4} + 29.68x_{9.4} + 13.95x_{10.4} \rightarrow max. \tag{18}$$

The second criterion is maximization of the total degree of the local population satisfaction:

$$f_2(x) = 0.11x_{1.1} + 0.40x_{2.1} + 0.70x_{3.1} + 0.30x_{4.2} + 0.80x_{5.2} + 0.30x_{6.3} +$$

$$+0.80x_{7.3} + 0.20x_{8.4} + 0.50x_{9.4} + 0.80x_{10.4} \rightarrow max. \tag{19}$$

The investments in new mining projects are limited to 75 million euro, which is why the limitation (11) will have a form:

$$12x_{1.1} + 16x_{2.1} + 21x_{3.1} + 24x_{4.2} + 28x_{5.2} + 18x_{6.3} + 27x_{7.3} + 16x_{8.4} + 19x_{9.4} + 25x_{10.4} \leq 75. \tag{20}$$

The first two mining project can be implemented if they have enough funding and they prove themselves to be expedient due to the used criteria. $j = 1$ mining project can be implemented in one of the three possible alternatives $i = 1, 2, 3$, so we should note down the limitation for it:

$$x_{1.1} + x_{2.1} + x_{3.1} \leq 1. \tag{21}$$

The second mining project $j = 2$ can be realized in one of the two options $i = 4, 5$. The limitation have the following form:

$$x_{4.2} + x_{5.2} \leq 1. \tag{22}$$

Mining projects $j = 3$ and $j = 4$ are obligatory, so the limitations of selecting options are the following:

$$x_{6.3} + x_{7.3} = 1, \tag{23}$$

$$x_{8.4} + x_{9.4} + x_{10.4} = 1. \tag{24}$$

All the target variables can be 0 or 1.

First, it is necessary to determine the limit (maximum and minimum) values of the criteria $f_1(x)$ and $f_2(x)$. Thus, the problems of minimization and maximization of each analyzed optimality criterion are solved separately taking into consideration above mentioned limitations. Table 13 shows the results of these calculations.

**Table 13.** Calculation of the optimal value of the target variables for alternatives of the mining projects implementation.

| Direction for Criterion Optimization | Optimal Value of the Criterion | Optimal Value of the Target Variables (Chosen Alternatives of the Mining Projects Implementation) | | | | | | | | | |
|---|---|---|---|---|---|---|---|---|---|---|---|
| | | Project 1 | | | Project 2 | | Project 3 | | Project 4 | | |
| | | $x_{1.1}$ | $x_{2.1}$ | $x_{3.1}$ | $x_{4.2}$ | $x_{5.2}$ | $x_{6.3}$ | $x_{7.3}$ | $x_{8.4}$ | $x_{9.4}$ | $x_{10.4}$ |
| $f_1(x) \rightarrow max$ | $f_1^{max} = 105.26$ | 1 | 0 | 0 | 1 | 0 | 1 | 0 | 1 | 0 | 0 |
| $f_1(x) \rightarrow min$ | $f_1^{min} = 24.37$ | 0 | 0 | 1 | 0 | 0 | 0 | 1 | 0 | 0 | 1 |
| $f_2(x) \rightarrow max$ | $f_2^{max} = 2.30$ | 0 | 0 | 1 | 0 | 0 | 0 | 1 | 0 | 0 | 1 |

| $f_2(x) \rightarrow \min$ | $f_2^{\min} = 0.91$ | 1 | 0 | 0 | 1 | 0 | 1 | 0 | 1 | 0 | 0 |
|---|---|---|---|---|---|---|---|---|---|---|---|

The analysis shows that in the case of the maximum value of the first criterion, those options of the mining projects implementation are chosen where only the lowest value of the second criterion is achieved. And vice versa, the maximum value of the second criterion leads to the selection of those alternatives, where the minimum value of the first criterion is obtained. The problem should be solved in accordance with two criteria at the same time in order to find a compromise solution. Based on the obtained limit values of the criteria, it is necessary to write down the scaling formulas (14) in numerical form:

$$F_1(x) = \frac{105.26 - f_1(x)}{105.26 - 24.37}, \tag{25}$$

$$F_2(x) = \frac{2.30 - f_2(x)}{2.30 - 0.91}. \tag{26}$$

Finally, the task of finding an optimal project maneuver has the following form:

$$z \geq \frac{105.26 - f_1(x)}{105.26 - 24.37}, \tag{27}$$

$$z \geq \frac{2.30 - f_2(x)}{2.30 - 0.91}. \tag{28}$$

With the limitations, (18)–(22) and criterion (15).

As a result of this problem solution, the value of the objective function is $z^* = 0.63$; thus, the deviation from the most optimal values of each criterion is 63%. Table 14 shows in detail the values of optimality criteria and the chosen options for mining projects implementation.

**Table 14.** Results of the problem solution in finding simultaneously a project maneuver in two categories with the same weight of optimality criteria.

| Compromise Value of the Criterion | Compromise Value for the Target Variables (Chosen Alternatives of the Mining Projects Implementation) | | | | | | | | | |
|---|---|---|---|---|---|---|---|---|---|---|
| | Project 1 | | | Project 2 | | Project 3 | | Project 4 | | |
| | $x_{1.1}$ | $x_{2.1}$ | $x_{3.1}$ | $x_{4.2}$ | $x_{5.2}$ | $x_{6.3}$ | $x_{7.3}$ | $x_{8.4}$ | $x_{9.4}$ | $x_{10.4}$ |
| $f_1^* = 53.97$ $f_2^* = 1.41$ | 1 | 0 | 0 | 0 | 0 | 0 | 1 | 0 | 1 | 0 |

According to the compromise solution, the first mining project is carried out on the basis of the first alternative, the second mining project is not implemented, the third one is executed with the use of the seventh option, and the fourth mining project is carried out with the use of the ninth alternative. Thus, it is possible to implement 1, 3, 4 mining projects which include more significant compensatory projects and are more profitable for the subsoil user (Figure 3).

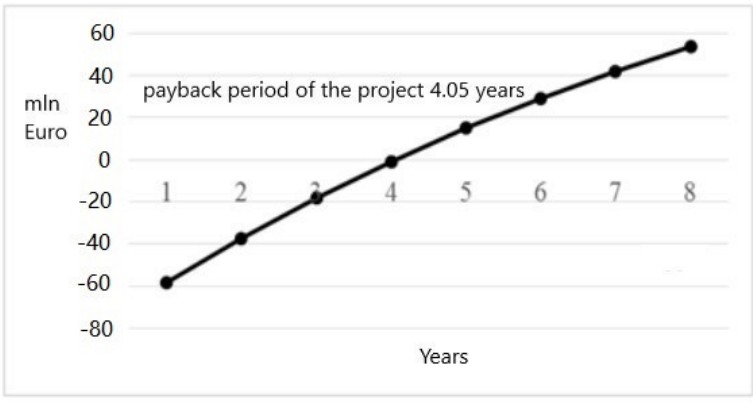

**Figure 3.** Net present value (NPV) and pay-back period with an optimal maneuver of the chosen mining projects.

The conducted project maneuver and corresponding options of project implementation allowed to satisfy the population up to 47% on average.

It is possible to increase the degree of the local population satisfaction but it leads to the deterioration of the economic characteristics of the mining projects. So, it is necessary to conduct a reanalysis of the optimal project maneuver with the help of an optimization model. There, weighting factors should be taken as optimality criteria, but the weight of such criterion as the maximization of the total degree of the local population satisfaction must be higher than the weight of the criterion of the maximization of the total net present value (NPV) during projects implementation. Table 15 represents the series of mathematical operations with continuous growth of the weight of the criterion—the maximization of the total degree of the local population satisfaction.

**Table 15.** Results of the problem solution in finding simultaneously a project maneuver in two categories with different weights of optimality criteria.

| Variant | Criteria Weights | Compromise Value for a Criterion | Compromise Value for the Target Variables (Chosen Alternatives of the Mining Projects Implementation) | | | | | | | | | |
|---|---|---|---|---|---|---|---|---|---|---|---|---|
| | | | Project 1 | | | Project 2 | | Project 3 | | Project 4 | | |
| | | | $x_{1.1}$ | $x_{2.1}$ | $x_{3.1}$ | $x_{4.2}$ | $x_{5.2}$ | $x_{6.3}$ | $x_{7.3}$ | $x_{8.4}$ | $x_{9.4}$ | $x_{10.4}$ |
| 1 | 0.3 | $f_1^* = 50.71$ | 0 | 0 | 0 | 0 | 1 | 1 | 0 | 0 | 0 | 1 |
| | 0.7 | $f_2^* = 1.90$ | | | | | | | | | | |
| 2 | 0.2 | $f_1^* = 40.11$ | 0 | 0 | 1 | 0 | 0 | 0 | 1 | 0 | 1 | 0 |
| | 0.8 | $f_2^* = 2.00$ | | | | | | | | | | |
| 3 | 0.1 | $f_1^* = 24.37$ | 0 | 0 | 1 | 0 | 0 | 0 | 1 | 0 | 0 | 1 |
| | 0.9 | $f_2^* = 2.30$ | | | | | | | | | | |

These calculations make it possible to determine the optimal project maneuver for each option. The growth of the total degree of the local population satisfaction is achieved by the reduction of the total net present value (NPV) during mining projects implementation. The reduction of the mining company's NPV has its limits which are connected with the company's financial strength, price risks, and sales volume. If this limit is 45.0 million euro, it would be well to settle on the first variant from

Table 15, as it corresponds to a fair compromise over the interests of a subsoil user and the indigenous peoples.

This approach allows to take into consideration not only interests of a mining company and the local population, but also makes it possible to assess the ecological danger in emergency situations during the mining [18] and to consider compensation for the environmental harm to the mineral exploitation economy [19]. These types of calculations can be an information base for the implementation of the State's environmental policy during the economic exploitation of the territory, not only at the State level but also at the mining company one [20].

## 4. Conclusions

The Arctic industrial development and the projects implementation of natural resources extraction affect the living conditions of the indigenous peoples and their traditional lands [21]. The mining activity in the Arctic impacts on climatic processes, on the quality of water resources, and on the health of local population. All these factors play significant role in the process of project maneuver selection making by mining company [22]. It chooses the alternative options for the technology application on the territory where the mining activity is planned or has been already occurring. The compensation projects are also the subject of the selection [23].

In the paper, the models for the selection of the options for alternative and compensatory projects and their criteria have been proposed. In our opinion, the criteria of such procedure can be: ensuring the health of the local population, providing local communities with high-quality drinking water, possible climatic changes due to development of mineral deposits (construction of roads, infrastructure), and the creation of new jobs and local employment. These issues are the subject of further research and affect the interests of the mining company and the local population in terms of finding the parity of economic welfare, environmental well-being, and preservation of the traditional culture.

To take into account the interests of the local population, a mining company can use various alternative technologies and projects from mining extension to abandoning development and transferring of the production to other deposits, etc.

The article formulates and resolves the task of procedures development for evaluating and selecting projects for diamond mining using the cases of subsoil users' activities in Yakutia. The main idea is to develop tools for projects implementation taking into account the needs and preferences of the local population. The numerical characteristics of alternative development projects for this territory are substantiated, including capital and operating costs, annual income, and the degree of satisfaction of the local population. This decision-making algorithm allows choosing the optimal project maneuver of the producing company when implementing compensation projects.

The calculation of the optimal project maneuver of a mining company is based on two criteria of optimality (net discounted income and payback period). The proposed approach solves the optimization problem. It allows to link and to harmonize the needs and interests of the mining company and the local population by choosing compensation projects based on maximizing the total degree of satisfaction of the indigenous peoples. The approach has a universal character. It can be used to justify various investment projects and allows determining a fair compromise between the interests of a mining company and the population of the region.

The idea of preparing this paper came from the numerous expeditions to the Arctic regions and meetings with indigenous peoples, mining companies, and local government. The motivation of the authors of the article is to improve modern methods of sustainable development management. The problem of the Arctic development is connected with the need to make this process more inclusive with active participation of indigenous peoples. As a result, the theory and practice of the Arctic development has got the tool to coordinate the interests of all stakeholders and to avoid the conflicts.

**Author Contributions:** Conceptualization, I.P. and A.N.; methodology, I.N.; software, A.N. and I.N.; validation, I.P., V.G. and I.N.; formal analysis, A.N.; investigation, V.G. and I.P.; resources, I.P.; data curation, A.N.; writing—original draft preparation, I.P. and V.G.; writing—review and editing, V.G.; visualization, V.G.; supervision, V.G.; project administration, V.G. and I.P. All authors have read and agreed to the published version of the manuscript.

**Funding:** This research received no external funding.

**Conflicts of Interest:** The authors declare no conflict of interest.

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
