# Peer review of "Sustainable Development of the Arctic Indigenous Communities: The Approach to Projects Optimization of Mining Company"

_sustainability, doi:10.3390/su12197963_

Round 1
Reviewer 1 Report
The paper presents the optimal decision choice for mining (diamonds, section 2). The exploitation of Tarsand in Alberta gathers indigenous communities. One base of the paper is the “hierarchy analysis method by T. Saaty”. The paper uses “alternative” for mining project and “compensatory” (Table 2). The paper ends with linear programming.
We notice that j is a mining project at line 207, and that j is a Variety of projects (understand as variable) in Table 2. “i” is a project.
ecpu equation (7) is difficult to understand, (от 0 до 1) line 250.
The second part of the paper use “maneuver” (line 260) before introducing NPV (line 329) which is defined later and it represents “Net present value” (line 371).
The logic of the paper is to assemble mathematical formulations and criteria presentation and discussion. We think that significant progress in the presentation is possible.
Reviewer 2 Report
Overall comments
The manuscript entitled “Sustainable development of the Arctic indigenous communities: the discus on industrial projects realization in traditional lands” by Andrey Novoselov et al. presents some interesting contents. As described in the manuscript, in the interest of indigenous people, it is crucial to perform a project maneuver during the development of the Arctic such that possible mining alternatives can be considered. However, the manuscript is difficult to comprehend with an overly complex structure because parts of the methodology and discussion are included in the results section. All the details about the methodology for the model development should be presented in Section 2, and the discussion should be separated from the results so that the paper has a logical flow. For better comprehension of the readers, the authors should also provide a flow chart of their model with an indication of the input and output parameters. In addition, some terminologies are not well defined in the manuscript. Climate change risks should also be explained because the Arctic region is strongly affected by climate change and also has an impact on it, for example, permafrost melting and drying of land due to severe warming; the development of diamond mining might enhance such risks to indigenous people. Overall, the authors need to include a schematic of the entire model flow in the methodology section, detail the results, provide a discussion, define specific terminologies at their first mention, add a geographical map, and mention climate change risks.
Major comments
(1) First of all, the results section includes details regarding the methodology and discussion. Please separate each section as methods, results, and discussion to create a logical flow. The entire model structure must be detailed in Section 2. Please add a flow chart of the model to include the input and output parameters. The flow chart could also include boxes summarizing the main results.
(2) Secondly, terminologies are not well defined in the manuscript. For instance, “ALROSA OJSC,” “subsoil user,” “NPV,” and “pay-back period” are not defined, and yet they are used as key words. The readers could become confused without these definitions. In particular, I cannot completely understand the term "subsoil user", which is used in many parts of the manuscript. Please define specific terms in the manuscript at their first mention.
(3) Thirdly, it is necessary to include a map of the diamond mines shown in Table 1. In addition, please point out the location of the Ebelyakh River on the map. Most readers will not be familiar with the geographical information of the Shakha Republic and Russian Arctic. In addition, please indicate the target region for the model application.
(4) In the Russian Arctic, the impact of global warming is larger than that of other low- to mid-latitude regions, and climate change affects the water and carbon cycles. For instance, Huan et al. (2015, doi: 10.1038/nclimate2837) and Suzuki et al. (2016, doi: 10.1080/01431161.2016.1165890) pointed out the expansion of dryland in the Russian high Arctic. The development of diamond mining in the Ebelyakh River watershed will accelerate hydrological changes and create a water resource problem for the indigenous people. It would be better to mention the potential impact on water resources in the discussion.
(5) Line 192: “According to the hierarchy analysis method by T. Saaty,...” Please add the complete citation here.
Reviewer 3 Report
Dear colleagues,
The title of the article looks really interesting and promising and the article would be highly valued but for:
- A great number of mistakes in English grammar and style made starting from the very title itself;
- Too much attention irrelevantly and unproportionally paid to the operational activities of the mining companies;
- The aim of the paper declared by you ("to work out a tool, algorithm and procedures for decision-making based on economic-mathimatical models developed by the authors") looking quite afar from the title of the article
- A lack of literature review on the problem of sustainable development of indigenous communities;
- The chapter "Material and Methods" comprising almost none of any description of material and methods used for the analysis and research;
- The qualitative and quantitative analysis of the collected data being absolutely erroneous because of the lack of relevance in the basic principle of any scientific classification that declares that any classification is to be based on a single characteristic of division. But to mix "reindeer herders, hunters, fisherman, pickers, gatherers, housewives" on the one hand which can refer to the characteristic "occupation" and, on the other hand, "young people and pensioners" that can refer to the characteristic (semantic field) "age" is an absolute flaw that downgrade the quality and relevance of the analysis results critically.
So I am sorry to say that I cannon see the work to be published in the present from without radical reconsideration.
Reviewer 4 Report
Thank you for your important contributions and thinking.
I have minor suggestions to improve readability and flow. These are:
Add that health and culture to features affected by mining projects – potentially in the Abstract but definitely in the Introduction and Conclusion
The section numbered ‘2. Materials and Methods’ reads more like a ‘Background’ section. It provides excellent material on context however does not include research methods information as such.
The section numbered ‘3. Results’ includes information about methods. In this section research methods information is included as well as findings/results. One important structural improvement is re-titling this to be ‘3. Methodology and Methods’ and ensuring all information in this is descriptive about methods. Then, on page 9 line 266, create a new title ‘4. Results’. The focus of the text at this point does become more on findings.
If material from page 9 line 266 becomes ‘4. Results’ some minor editing of wording will be required for flow. Doing this will also reduce the length of the existing section 3.
Information about methodology is needed to contextualise the methods. Provide brief explanations of ‘What are the Disciplines of the authors?’, ‘What is the interest of the authors in this research including motivations to do this work?’ ‘What relationships and experience do the authors have with Indigenous people, in order to appropriately present this work, which could have profound impacts on their lives?’ These are all important elements of methodology, which frame methods. These are important given this is a methods-driven paper advocating for uptake of decision-making tools for the future.
Prioritisation: further to the above point about methodology, include a brief explanation of culturally appropriate ways of engaging Indigenous people in priority criteria setting – how the priority criteria are set. This could go on page 5 line 169 approximately
A reference for T. Saaty page 6 line 192 will strengthen this point.
Page 6 line 199 – Do you have any suggestions for how judgements could be clarified by experts? Consider adding a sentence or two here, and also citations to material that could be useful to the model, its use and therefore to the reader.
Page 11 – Add information relevant to this material about how populations were identified – how were they divided in groups and who by? What power did Indigenous people have in this identification, and if none, could they, and how? This is important for others to use.
Consider the addition of a section ‘5. Limitations’. Include in this the extent of participation of Indigenous peoples in the research and interpretation of findings.
Add to the Conclusions section a brief reflection on the health, wellbeing and environmental impacts of mining.
Language
Minor editing to improve English written expression is required throughout; the paper should not be published without this.
Check – does “ALROSA” require the double quotation marks? Ensure the acronym is explained first before its first use.
Consider ‘agreement’ rather than ‘compromise’ page 1 line 18
Avoid use of ‘etc’ in Abstract – too casual
Page 2 line 48 ‘natural’ not ‘native’ environment
Round 2
Reviewer 1 Report
The quality of the revised paper flow improved drasticaly. The following notations correspond to clear concepts : j , i , Pij, building together a hierachical structure.
The definitions of the following notations x_ij (line 258), x, x_1 x_2 x_3 (table 12), f , f_1 (equation 9), f_2 (equation 10) and F_1 F_2 (equations 14, 23-24) can be found in the paper. Sometimes it appears two times, for example z (equations 15 and 25) . The reader could understand that a mathematical programming starts only in the section 3, ending with the compromise optimization F1, F2. It appears as Pareto domain in the first part of the paper.
Line 405 mentions Table 11 on the fence of pages 13-14.
Edit column names Table 9.
Some similarities with tar sand exploitation in Alberta could be made (water ressource, impact on local population).
The success of the paper is to study theory and practice of Arctic development.
Human health dimension increases in the revised paper.
Check constraints (9-11) begining line 293
Coefficients of equations (16,17) belong to the surrounding Table 11.
Caption Table 12 on the fence pages 14-15
Check (12) line 388
Reference list is complete
The manuscript has been significantly improved and now warrants publication in Sustainability.
Reviewer 2 Report
Major comments
The paper titled “Sustainable development of the Arctic indigenous communities: the discus on industrial projects realization in traditional lands” by Andrey Novoselov et al. discusses the development of indigenous communities in the Russian Arctic based on the optimization of projects of mining companies. This paper provides important information that will contribute to how we can carry out the sustainable development of Arctic resources, which will be intensified by the thawing of permafrost and the ease of underground development.
Reviewer 3 Report
Dear colleagues
Since you have taken quite an effort to abide by what was mentioned in the previous review and it is obvious that the article has undergone dramatical changes I have no further objection to it being published.
Thank you and kind regards.
